# Synthesis of Polyacids by Copolymerization of L-Lactide with MTC-COOH Using Zn[(acac)(L)H_2_O] Complex as an Initiator

**DOI:** 10.3390/polym14030503

**Published:** 2022-01-27

**Authors:** Joanna Jaworska, Michał Sobota, Małgorzata Pastusiak, Michał Kawalec, Henryk Janeczek, Piotr Rychter, Kamila Lewicka, Piotr Dobrzyński

**Affiliations:** 1Centre of Polymer and Carbon Materials, Polish Academy of Sciences, 34 Curie-Sklodowskiej Str., 41-819 Zabrze, Poland; jjaworska@cmpw-pan.edu.pl (J.J.); msobota@cmpw-pan.edu.pl (M.S.); mpastusiak@cmpw-pan.edu.pl (M.P.); mkawalec@cmpw-pan.edu.pl (M.K.); hjaneczek@cmpw-pan.edu.pl (H.J.); 2Faculty of Science and Technology, Jan Dlugosz University in Czestochowa, 13/15 Armii Krajowej Av., 42-200 Czestochowa, Poland; p.rychter@ujd.edu.pl (P.R.); k.lewicka@ujd.edu.pl (K.L.)

**Keywords:** biodegradable polymers, functional polymers, ring opening polymerization, coordination polymerization, chelate complexes

## Abstract

This work presents the results of research on the preparation of bioresorbable functional polyestercarbonates containing side carboxyl groups. These copolymers were synthesized in two ways: the classic two-step process involving the copolymerization of L-lactide and a cyclic carbonate containing a blocked side carboxylate group in the form of a benzyl ester (MTC-Bz) and its subsequent deprotection, and a new way involving the one-step copolymerization of L-lactide with this same carbonate, but containing an unprotected carboxyl group (MTC-COOH). Both reactions were carried out under identical conditions in the melt, using a specially selected zinc chelate complex, with Zn [(acac)(L)H_2_O] (where: L-N-(pyridin-4-ylmethylene) phenylalaninate ligand) as an initiator. The differences in the kinetics of both reactions and their courses were pictured. The reactivity of the MTC-COOH monomer without a blocking group in the studied co-polymerization was much higher, even slightly higher than L-lactide, which allowed the practically complete conversion of the comonomers in a much shorter time. The basic final properties of the obtained copolymers and the microstructures of their chains were determined. The single-step synthesis of biodegradable polyacids was much simpler. Contrary to the conventional method, this made it possible to obtain copolymers containing all carbonate units with carboxyl groups, without even traces of the heavy metals used in the deprotection of the carboxyl groups, the presence of which is known to be very difficult to completely remove from the copolymers obtained in the two-step process.

## 1. Introduction

At present, there is an incessantly growing interest in the issue of the synthesis and application of biodegradable polymers [1,2]. Among the increasing number of materials of this type, special biodegradable polymers with embedded functional group currently represent especially attractive materials for biomedical applications [3,4,5,6].

Polymers containing functional carboxylic groups are appealing due to their chemical activity and their ability to form covalent bonds with reactive groups of active agents including drugs. In this respect, such functional polymers represent very promising materials for use as carriers of active agents in the drug and protein delivery systems that are currently being successfully used in pharmacy and cosmetology. Due to the presence of carboxylic groups in such polymers, it is possible to modify their physicochemical properties (via the modification of hydrophilicity and protein binding), giving them advantages over the conventional aliphatic polyesters and polycarbonates. Thanks to these properties, such polymers may be used in tissue engineering for the formation of 3D scaffolds for application in cell cultures.

As copolymers that are obtained via the copolymerization of lactides, lactones with cyclic carbonates containing functional pendant carboxylic groups are great candidates that meet the criteria described above. Chain growth polymerization makes it possible to easily obtain high molecular polymers and copolymers with high purity, which is especially required for biomaterials. Cyclic carbonates containing carboxylic groups may be used as monomers in the co-polymerization reaction via ROP with lactides and lactones, resulting in the synthesis of copolymers containing embedded functional groups [7].

With the proper selection of the reaction mixture’s composition, including the molar ratio of lactide to cyclic carbonates, it is possible to control the number of functional groups in the obtained copolymer chain. As consequence, this is reflected in the monitoring of polymers’ hydrophilicity, rate degradation, or solubility in water. Unfortunately, non-blocked carboxylic groups in cyclic carbonates are significant hindrances during ring-opening polymerization with lactides and lactones. The vast majority of commonly used initiators easily react with carboxylic groups, leading to the very rapid termination of chain growth. For this reason, before polymerization, it is necessary to protect the carboxylic groups of the monomers by blocking them; this can take place, for example, through the creation of esters via a reaction with benzyl chloroformate [8]. Unfortunately, the main disadvantage of such a reaction is the fact that after obtaining the final polymer, protected carboxylic groups have to be unblocked. Moreover, the polymerization of monomers containing pendant ester groups may lead to a parallel reaction to the main chain propagation; specifically, it may lead to transesterification being directed toward this group; therefore, it is difficult to obtain a linear structure of the final polymer chain. Due to this phenomenon, widely branched and crosslinked polymers are very often obtained. The degree of such transesterification is dependent on the type of monomer, the initiator, and the reaction temperature. Therefore, to avoid transesterification reactions, it is interesting to look at the results of the research on the ROP polymerization of lactides with the participation of zinc initiators containing heteroleptic ligands, without co-initiators. [9]. Thus, with the appropriate selection of these features, the contents of the cross-linked fraction can be significantly reduced [10].

Contrary to many earlier opinions, it was found that the selection of a proper initiator allows for the ROP of the derivatives of cyclic carbonates containing carboxylic groups without blocking these functional groups. Al-Azemi et al. [11] revealed that by using lipase (Novozym-435 and lipase AK) as a catalyst in the copolymerization of trimethylene carbonate (TMC) with the cyclic carbonate-containing carboxylic group MTC-COOH (5-methyl-2-oxo-1,3-dioxane-5-carboxylic acid), it is possible to obtain a copolymer with the designed composition and a linear chain structure. The obtained yield of ROP was dependent on the amount of MTC-COOH in the reaction mixture and was ca 70 and 90%, respectively. Using lipase, in a one-step reaction, the terpolymers of MTC-COOH with TMC and ADMC (6,14-dimethyl-1,3,9,11-tetraoxa-6,14-diaza-cyclohexadecane-2,10-dione) were also obtained [12]. Importantly, it was also found that coordination complexes such as zirconium IV acetylacetonate Zr(acac)_4_ may be successfully used as initiators in the ROP of monomers with active carboxylic groups, such as MTC-COOH. This phenomenon was previously explained in detail. Briefly, Zr(acac)_4_ was found to initiate ROP differently when compared to popular catalysts such as stan-nous (II) octoate or some metal alkoxides, and did not participate in the reaction with carboxylate groups [13]. The obtained results suggested that the selected complexes, which were obtained through the exchange of some acetylacetonate ligands via reactions with proton donor compounds, should be even better initiators of such reactions.

The present work aimed to study the course of the copolymerization of L-lactide and cyclic MTC-COOH, using in one- and two-steps reactions for comparison, and to characterize the obtained copolymers. Initially, as an initiator, we tested zirconium, molybdenum, and zinc compounds (the previous selection was related to the low toxicity of these compounds) obtained in the ligand exchange reaction of appropriate acetylacetonates and Schiff bases. Zn[(acac)(L)(H_2_O)] was finally selected as the optimal compound. This complex was obtained via the ligand exchange of Zn[(acac)_2_(H_2_O)] with the Schiff base, which was synthesized in a reaction with phenylalanine [14]. It demonstrated higher initiation activity when compared to zirconium (IV) acetylacetonate, while, at the same time, displaying antibacterial activity with a lack of cytotoxicity against fibroblasts. The zinc complex used in this study initiated the growth of the polylactide chain via a similar mechanism, which was observed during the use of the Zr(acac)_4_ initiator [15,16].

## 2. Materials and Methods

### 2.1. Materials

The monomer, L-lactide (LA) (Huizhou Foryou Medical Devices Co., Ltd., Huizhou, China), was purified by means of recrystallization from anhydrous ethyl acetate and then dried in a vacuum oven at room temperature until a constant weight was reached.

Zinc (II) acetylacetonate monohydrate (Avantor Performance Materials Poland S.A., Gliwice, Poland), Pd/C palladium catalyst (Merck Sp. z o.o., Warszawa, Poland), N,N dimethylformamide (99.8%) (Chempur, Piekary Slaskie, Poland), ethyl chloroformate (97%), (Alfa Aesar, Heysham, Lancashire, United Kingdom), potassium hydroxide (99.8%) (Merck Sp. z o.o., Warszawa, Poland), L-phenylalanine, 4-pyridine carboxaldehyde, methanol anhydrous (99.8%), benzene anhydrous (99.8%), chloro-form anhydrous (99%), tetrahydrofuran anhydrous (99.9%), 2,2-Bis(hydroxymethyl) propionic acid (bis-MPA) (98%), and benzyl bromide (98%) were purchased from Merck Sp. z o.o., Warszawa, Poland. All these chemicals were used as received.

### 2.2. Synthesis of Functional Carbonate Monomers (Benzyl 5-Methyl-2-oxo-1,3-dioxane-5-carboxylate and 5-Methyl-2-oxo-1,3-dioxane-5-carboxylic Acid)

5-Methyl-2-oxo-1,3-dioxane-5-carboxylic acid (MTC-COOH) was synthesized according to a previously described method [17,18]; in detail, bis-MPA (270 g, 2.013 mol, 1 eq.) was dissolved in 1.3 dm^3^ of DMF and KOH (135.537 g, 2.416 mol, 1.2 eq.) was added. A clear solution was obtained after the mixture was heated to 100 °C. Next, benzyl bromide (287 cm^3^, 413.134 g, 2.416 mol, 1.2 eq.) was added and the reaction was carried out overnight at 100 °C. Then, after DMF had been stripped off under reduced pressure, the nonvolatile compounds were dissolved in 3.0 dm^3^ of dichloromethane and the organic solution was washed 3 times with 500 cm^3^ (each wash) of aq. dist. followed by drying over anhydrous MgSO_4_. After MgSO_4_ had been filtered off and the solvent had been stripped off, the residual solid was recrystallized from toluene to yield white crystals of benzyl 2,2-bis(methylol) propionate, which were dried under vacuum (T = 50 °C) to constant weight. Yield = 61%. Next, under nitrogen flow, triethylamine (161 cm^3^, 117.18 g, 1.158 mol, 3 eq.) was introduced dropwise into a solution composed of benzyl 2,2-bis(methylol) propionate (86.5 g, 0.386 mol, 1 eq.) and ethyl chloroformate (110 cm^3^, 125.72 g, 1.158 mol, 3 eq.) in 1500 cm^3^ of chloro-form CHCl_3_, which was thermostated in an ice/water bath. Then, the cooling bath was removed and the reaction was carried out overnight. Next, the mixture was washed with 2 × 150 cm^3^ 1M HCl_aq_, 150 cm^3^ saturated aqueous solution of NaHCO_3_, 150 cm^3^ brine, and finally 2 × 50 cm^3^ aq. dist. The organic phase was dried over anhydrous MgSO_4_. Next, MgSO_4_ was filtered off, CHCl_3_ stripped off, and the nonvolatile residue was recrystallized from toluene to yield white crystals of benzyl 5-methyl-2-oxo-1,3-dioxane-5-carboxylate (MTC-Bz). Yield = 81%. This compound was used in copolymerization with L-lactide.

After the addition of 1g of Pd/C into the solution of benzyl 5-methyl-2-oxo-1,3-dioxane-5-carboxylate (10.0 g) in 150 cm^3^ of ethyl acetate or THF, the reaction with hydrogen was conducted in a pressure hydrogenation reactor at 3 atm (Paar Shaker Hydrogenation Apparatus 3900, Parr Instrument Company, Moline, IL, USA). The hydrogenation process was carried out over the course of one day, and the resulting material was dissolved in 100 cm^3^ of THF and filtered several times with the help of a Buchner funnel equipped with diatomaceous earth and cellulose pulp. Next, the solvents were stripped off and the resulting white crystals of 5-methyl-2-oxo-1,3-dioxane-5-carboxylic acid (MTC-COOH) were dried under vacuum to constant weight. Yield = 82%. In this form, without further purification, the MTC-COOH crystals were used in copolymerization reactions with L-lactide. This same method was used during the deprotection of MTC-Bz/L-lactide copolymers but the process was extended to 3 days.

### 2.3. General Procedure for the Synthesis of the Zinc Initiator; Zn[(acac)(L)H_2_O] (Where: L-N-(2-Pyridin-4-ylethylidene) Phenylalaninate Ligand)

A detailed description of the synthesis of the zinc (II) monohydrate complex, containing acetylacetonate and a second chelating ligand, as being derivative of the Schiff base (Figure 1) was given in our previous paper [14]. This complex has already been used as an initiator in the copolymerization of L-lactide. The initiator was obtained via the ligand acetylacetonate exchange reaction of the Zn[(acac)_2_ H_2_O] complex with the Schiff base, which was synthesized through the condensation of L-phenylalanine with 4-Pyridinecarboxaldehyde. The final product was purified through double crystallization from methanol.

### 2.4. Copolymerization Procedures

A copolymerization of MTC-Bz with L-La and MTC-COOH with L-La was carried out in bulk at 120 °C. Zn[(acac)(L)H_2_O] was used as an initiator of this reaction with the initiator/monomer ratio (I/M) of 1/600. An exemplary procedure to obtain the copolymer (MTC-Bz/L-lactide 50) can be described as follows: Quantities of 5 g (0.02 mol) of MTC-Bz, 2.9 g (0.02 mol) of L-lactide, and 0.017 g (6.7 × 10^−5^ mol) Zn[(acac)(L) (H_2_O)] were introduced into a glass reaction vessel with 50 cm^3^ capacity, connected to a vacuum line, and dried argon. The reactor content was degassed and stirred under argon at 120 °C for 48 or 72 h. The product was dissolved in chloroform, precipitated with cold methanol, and then dried in a vacuum oven to constant weight.

### 2.5. Measurements

The conversion of the reaction, the composition of the copolymers, and the microstructure of the chains (the average length of microblocks, and the randomness of the copolymer chain (R)) were determined using NMR spectroscopy. The ^1^H NMR spectra were recorded at 600 MHz with a Bruker Avance II TM (Bruker BioSpin GmbH, Rheinstetten, Germany) at 25 °C. Dried DMSO-d6 was used as a solvent, and tetramethylsilane was applied as the internal standard. The spectra were obtained with 64 scans, a 2.65 s acquisition time, and an 11 µs pulse width. The mean length of the microblocks and the randomization factor of the R copolymer chain were determined from the analysis of the ^13^C NMR spectrum, and these were recorded at 150 MHz at 40 °C and 80 °C with the same spectrometer. Samples of 15 mg were dissolved in 0.4 mL of deuterated DMSO-d6. The acquisition time was 0.9 s, the pulse width was 9.4 ms, the delay between pulses was 2 s, and the spectral width was 36,000 Hz.

The number-average and weight-average molar masses of the oligomers were determined by gel permeation chromatography with a Viscotek RImax chromatograph (Malvern Panalytical Ltd., Malvern, UK). Chloroform was used as the eluent, and the temperature and flow rate were 35 °C and 1 mL/min, respectively. Two PL Mixed E columns were used, with a Viscotek model 3580 refractive index detector and an injection volume equal to 100 µL. 

The transition temperature T**_g_**, melting temperature T**_m_** and melting enthalpy ΔH**_m_** were examined by differential scanning calorimetry (DSC) with a TA DSC Q2000 apparatus (TA Instruments, New Castle, DE, USA) according to the ASTM E 1356–08 standard. The instrument was calibrated with high-purity indium. The samples were heated and cooled at a heating rate of 20 °C/min from −50 °C to 220 °C. All of the experiments were performed with a nitrogen flow rate of 50 mL/min.

## 3. Results

### 3.1. Copolymerization of L-Lactide with 5-Methyl-2-oxo-1,3-dioxane-5-carboxylate (MTC-Bz)

At the first stage of the study, a series of copolymerization reactions of L-lactide with the previously obtained MTC-Bz were conducted at various molar ratios of both comonomers. The reactions were conducted in bulk at a temperature 120 °C using a zinc complex, Zn[(acac)(L) H_2_O], as an ROP initiator (Figure 2). There were a few reasons why this type of initiator was chosen. First of all, it was already proven to have show initiation activity in the ROP of L-lactide, much lower catalytic activity in transesterification reactions when compared to zircon-based compounds, a lack of cytotoxicity, and relatively strong antibacterial and antifungal activity [14].

The obtained copolymerization products were analyzed via the NMR technique. The spectra of the MTC-Bz and LA homopolymers as well as the spectra of copolymers that differed in their contents of comonomers (30/70, 50/50, 70/30) were compared for detailed analysis. Previously reported results devoted to the copolymerization of trimethylene carbonate with MTC-Bz were very helpful during the assignation of signals in the NMR spectra of the synthesized compounds [10] or the synthesis of aliphatic poly(ester-carbonate) with folic acid functionalization [19]. A representative ^1^H NMR spectrum of equimolar copolymer is presented in Figure 1.

Based on the ^1^H NMR spectra, the conversion of monomers and copolymer composition was calculated. For this purpose, methyl proton (CH_3_) signals from MTC-Bz units and methyl proton signals (CH_3_) in LA units were integrated (3H for MTC-Bz, 6H for LA). ^13^C NMR spectra were used to determine the copolymer chain’s microstructure. The sample spectrum of the MTC-Bz/LA copolymer (50/50) with the assignation of signals is presented in Figure 2.

Due to the copolymerization process, the majority of signals recorded on the ^13^C NMR spectra were split, which suggests that they were sensitive to the arrangement of the sequence in the polymer chain (Figure 3).

Moreover, the splitting observed in the different regions of the ^13^C NMR spectrum of the MTC-Bz homopolymer resulted from the presence of an asymmetric carbon atom in the chain. The carbonyl carbon signal of the lactide units was used to calculate the average length of the blocks. These signals split into two lines, and their dyad sequences are shown in Figure 4.

The average lengths of the microblocks were determined on the basis of the following equations:(1)$$L{\mathrm{LL}}_{\;}=\frac{\left[\mathrm{LLLL}\right]+\left[\mathrm{MLL}\right]}{\left[\mathrm{MLL}\right]};$$
(2)$$LM_{\;}=L_{\mathrm{LL}}\;\times \;\frac{F_{M}}{1-F_{M}}$$
where [LLLL], [MLL] is a concentration of comonomeric sequences and FM is a percentage content of M units

The values of the average lengths of the microblocks obtained in calculations (1) and (2), expressed in lactidyl or carbonate units, are presented in Table 1.

The final chain structure of the synthesized copolymers was dependent not only on the reactivity of the monomers in the studied reactions but also on the occurrence of the parallel intermolecular transesterification reaction. The randomness coefficient of the copolymer chain was chosen as a parameter describing the rate of the transesterification reaction. This parameter was already used in our previous reports [20]. The randomization ratio of copolymer chains (R) was determined with the following Equation (3):(3)$$R=\frac{L_{LL}^{R}}{L_{LL}}=\frac{L_{M}^{R}}{L_{M}}$$
where $L_{LL}^{R}$ and $L_{M}^{R}$ are the average lengths of the lactidyl and MTC blocks, respectively, in chains with completely random contributions of carbonate and lactidyl units. These may be determined from sequences (4) and (5):(4)$$L_{LL}^{R}=\frac{k^{\prime }+1}{2k\prime }\;\mathrm{and}\;L_{M}^{R}=k^{\prime }+1$$
(5)$$\mathrm{and};\;k^{\prime }=\frac{1-F_{L}}{F_{L}}$$
(6)$$F_{L}=\frac{2F_{LL}}{F_{LL+1}}—\mathrm{the}\;\mathrm{molar}\;\mathrm{fraction}\;\mathrm{of}\;\mathrm{the}\;\mathrm{lactidyl}\;\mathrm{units}\;\mathrm{in}\;\mathrm{the}\;\mathrm{copolymer},$$
where $F_{LL}$ is the molar fraction of LL-lactidyl.

Thus, if the value of parameter R is close to 1, the copolymers demonstrate the statistical structure of the main chain. This is caused by the fact that during the synthesis of these copolymers, intermolecular transesterification reactions occur, which result in decreases in the average lengths of the lactidyl LL and carbonate MTC microblocks as opposed to the length only being a result of the differences in both comonomers’ reactivities. After ca 3–4 days of copolymerization reaction, almost 100% conversion of the monomers was achieved (Table 1) and, due to this fact, the composition of the obtained copolymers was very close to the composition of the initial mixture used in the reaction. As determined by GPC chromatography, the average values of the molecular weights of the synthesized copolymers were much lower than those theoretically calculated based on the amounts of initiator used. Since polystyrene standards were used during the calculations, the obtained results should be interpreted as rough approximations. A higher amount of MTC-Bz was taken to the reaction mixture, and then a significantly higher dispersion of the copolymers’ molecular mass was noticed (Figure 5). The copolymer with 70% MTC-Bz units (Table 1. Entry 5) demonstrated the formation of high molecular fractions, which caused a simultaneous increase in mass dispersion (Đ) to a value above 70. The obtained copolymer contained fraction that was insoluble in chloroform. In the case of the homopolymerization of MTC-Bz (Table 1, entry 6), the obtained polymer was practically insoluble in chloroform and THF, and formed a microgel. The growing amount of carbonate sequences in the copolymer chain caused an increase in the chain randomness rate (R). Thus, when there was a large number of carbonate groups in the copolymer chain, a high transesterification degree was noticed (Table 1, No. 5). In this particular case, the formation of the crosslinked fraction of the insoluble microgel was also observed. This phenomenon was undoubtedly connected with the processed reaction that occurred in parallel to the main propagation of the copolymer chain, namely the intramolecular transesterification that was directed to the ester bonds of the benzyl side groups. Reactions of this type cause the formation of a high molecular fraction of agglomerates with a strongly branched chain structure and the creation of low molecular fractions at the same time. This phenomenon could be confirmed via the GPC analysis (Figure 5C). In addition, an analogous phenomenon was already observed during the copolymerization of ethyl 5-methyl-2-oxo-1,3-dioxane-5-carboxylate (MTC-Et) with trimethylene carbonate (TMC) initiated with Zn[(acac)_2_ H_2_O]. This process was much more intensive than that described in this work, as evidenced by the final dispersion of the molecular mass reaching a Đ value of more than 100 [21]. The growth of the amount of carbonate units in the obtained copolymers caused a decrease in their glass transition (T_g_); however, this decrease was significantly slower than expected based on the calculation using the Fox equation (Table 1). The reason for this phenomenon was the stiffening of the copolymer chain caused by an observed increase in the cross-linked fraction with the enhancement of the number of MTC-Bz units in the copolymer as well as the growth of the number of large benzyl substituents that mainly interacted, in an intermolecular manner, with the ester segments of the chain. All copolymers demonstrated semi-crystallinity. Depending on the copolymer composition, ordered phases were formed by longer lactidyl (for copolymer containing 70% of lactidyl units at the average length of this microblock about 6 lactidyl units) or carbonates (for copolymers containing 50 and 70% carbonate units) microblocks.

### 3.2. Obtaining Polyacides, Poly (L-Lactide-co-MTC-COOH), via the Deprotection of Poly (L-Lactide-co-MTC-Bz)

The obtained copolymers, after being dissolved in THF, were subjected to hydrogenation, at a pressure of 3 atm, in the presence of a palladium catalyst (Figure 3). The product, after purification from the catalyst and distillation of the solvent, was subjected to further tests.

^1^H NMR spectra were used to confirm the deprotection ratio and to evaluate the composition of the obtained mixture of the copolymer with a deprotected carboxyl group (MTC-Bz/MTC-COOH/LA). Figure 6 presents the ^1^H NMR exemplary spectra obtained for copolymers containing about 70% lactidyl unit; the initial MTC-Bz/LA copolymer (A) and the final deprotected MTC-Bz/MTC-COOH/LA (B) are shown. The partial reduction in the number of protecting benzyl groups was confirmed based on the signal integrations: the integration value of the -C_6_H_5_ signal was reduced after deprotection (the -CH_3_ signal was chosen as a reference). A weak but wide signal, observed at 10–15 ppm, appeared to be new, and this signal was assigned to unblocked carboxylic groups.

Table 2 presents the comonomer units’ content in the mixture after deprotection. Calculations were made based on the ^1^H NMR spectra. It can be seen that deprotections were only partial—7 to 33% of MTC-Bz remained, depending on the starting copolymer composition. Despite the longer duration of the reaction (3 days), the large amount of catalyst, and the use of pressure, this process ran very slowly. In the case of the copolymer that contained the highest amount of carbonate units (Table 2, 4H), it failed to unblock even half of the carboxylic groups, probably due to the very low solubility of the obtained copolymer.

After the deprotection of the carboxyl groups, the average lengths of the lactidyl and carbonate chain segments and the chain randomness remained essentially unchanged compared to the starting L-LA/MTC-Bz copolymers (Table 1, Table 2). These calculations were made using the ^13^C NMR spectra (Figure 7) and the method described in the previous section. Based on the analysis of the GPC results, a slight decrease in the set average molecular weights was noted (Table 2), which may be explained by the lower molecular weight of the repeating unit of this copolymer after the releasing of the toluene, but also by the degradation of the main chain ester bonds that occurred during the hydrogenation process. However, the determined molecular weights should be treated as estimates, especially for copolymers with higher contents of carbonate units, due to the differences that occur during the separation process on the column packing due to the presence of active side carboxyl groups. The glass transition temperatures of all copolymers were similar; they were about 30 °C. Nevertheless, it can be seen that the contents of the pendant carboxyl groups, which made hydrogen bonding possible, determined the increases in the glass transition temperature. Copolymers containing longer carbonate microblocks showed semi crystallinity (Table 2, no. 3H, 4H), probably due to the presence of ordered regions containing carboxyl groups.

### 3.3. Direct One Step Synthesis of Copolymer of MTC-COOH with LA

MTC-COOH was directly copolymerized with LA in bulk (Figure 4). Because MTC-COOH is a monomer with a high melting point (166 °C), this process was carried out at a slightly higher melting temperature (130–135 °C) than it was in the earlier copolymerization of La with MTC-Bz.

A series of samples with different compositions was obtained and subjected to NMR tests (Table 3). As was also the case for the previously tested samples, ^1^H NMR copolymer spectra (Figure 8) were used to calculate the comonomer unit contents, which are presented in Table 3. The data relating to the microstructure of the copolymer chain (average micro block length, randomness) were determined with ^13^C NMR measurements (Figure 9) using the described signal assignments.

NMR measurements confirmed the expected composition of the copolymers as well as the presence of carboxyl groups in the chain. On the proton spectrum, a broad signal between 12 and 13.5 ppm, characteristic for protons of acidic groups, was noted, while on ^13^C, there was a signal that was typical for carbonyl carbon: HOC = O. The obtained spectra were very similar to the analogous of the copolymers synthesized in the two-step process.

During the conducted reaction, high conversion of comonomers was achieved within 24 h, irrespective of the starting composition of the reaction mixture, and thus, this occurred in a much shorter time than in the case of copolymerization of lactide with TMC-Bz. The average length of the micro blocks of the chain of the obtained copolymer was slightly shorter compared to the analogous La/MTC-Bz copolymers prepared by reaction with protected carboxyl groups. The copolymer chain showed a more randomized microstructure (with a R value of about 0.7), even though the differences in the reactivity of the comonomers were slightly greater (see Section 3.4). These parameters indicate a greater intensity of intermolecular transesterification and an increased role of this phenomenon in shaping the final structure of the copolymer chain. Nevertheless, only the copolymer containing the highest amount of carbonate units contained a small gel fraction when dissolved in chloroform or THF. This proves that the contribution of the transesterification reaction was low, thus enabling cross-linking to take place through the attack of the side carboxyl groups on the ester bonds of the polyestercarbonates chains. The glass transition temperatures slightly and proportionally increased, from 41 °C to nearly 50 °C, with the content of carbonate units in the copolymer. Despite the presence of long flexible chain sequences built of carbonate units, the glass transition temperature was relatively high. The main reason was the strong intermolecular interactions between the hydrogen of the carboxyl group and the oxygen of the ester bonds. The temperatures were higher as compared to the glass transition temperatures of analogous copolymers obtained by deprotecting the pendant carboxyl groups; this was due to the higher number of these groups present in the chain as well as their higher average molecular weight. The copolymers, which contained more than 30 mol% of carbonate units, showed a degree of semi-crystallinity that was similar to the analogous La/MTC-COOH/MTC-Bz copolymers that were described previously. In the case of the equimolar copolymer La/MTC-COOH, the heat of the fusion of the carbonate units’ region was lower, which was most likely related to the shorter average lengths of these segments. The melting points were very similar, amounting to over 100 °C. Copolymers containing a predominant amount of lactidyl units, due to their insufficient, overly short sequence length, did not form semicrystalline regions.

### 3.4. Comparison of the Copolymerization Course of Equimolar Amounts of L-Lactide with MTC-Bz and L-Lactide with MTC-COOH

In order to study the course of both copolymerizations, changes in the conversion ratios of monomers, depending on the reaction times and the relationships between monomer conversions, were analyzed. The reactions were carried out under the same conditions in the melt. Calculation of the comonomers’ conversion ratio was carried out on the basis of the ^1^H NMR analysis. The MTC-COOH/L-lactide copolymerization was much faster, and after about 24 h, 80% conversion was obtained. In the case of the second reaction, using a carboxyl-protected carbonate as a monomer, a similar conversion was obtained only after about 50 h (Figure 10A). The complete monomer conversion required 96h. During the observation of changes in the ratio of conversion of both comonomers (Figure 10B), along with the progress of the reaction, it was shown that during the copolymerization of L-lactide with MTC-COOH, the cyclic carbonate was slightly more active.

In the case of the copolymerization of L-lactide with MTC-Bz, the difference in the reactivity of the monomers was even smaller. The L-lactide was more responsive. The lower reactivity of the cyclic carbonate with the protected carboxyl group was most likely related to a steric effect resulting from the presence of a large benzyl side group.

## 4. Conclusions

It was shown that it is possible to perform the copolymerization of cyclic carbonates containing an unprotected carboxyl group with lactide according to the coordination-initiated ROP mechanism. However, it is necessary to select an initiating complex that, under the conditions of the reaction, will not react with this side group. Such a complex is the Zn [(acac)(L)H_2_O] chelate zinc complex. This one-step method of synthesizing biodegradable polyacids, compared to the conventional method of obtaining this type of copolymer used thus far, has great advantages. It is much easier to carry out, and makes it possible to obtain copolymers containing only carbonate units with pendant carboxyl groups. Such copolymers are practically impossible to obtain in a two-step process due to the problems with the full deprotection of these groups. It also seems that this is the only way to be able to obtain such biocompatible copolymers without traces of the heavy metals used as catalysts during the deprotection process. The reactivity of the cyclic carbonate monomer without a carboxyl blocking substituent is much higher, which makes it possible to achieve the practically complete conversion of the comonomers in a much shorter time. Comparing the properties of the MTC-COOH/L-lactide copolymers and the courses of their reactions with the analogous copolymers that were also obtained in a one-step enzymatic process [11], significantly higher degrees of conversion of both monomers (95–97% for the complex initiator; 72–92% for the reaction with an enzyme) as well as higher average molecular weights (M_w_) (32,000–49,000 g/mol for copolymers obtained with zinc complex; 7800–9000 g/mol for copolymers obtained with lipase) were visible.

Due to the high stability of the used initiator, the associated ease of storage, the good solubility in the reaction mixture, and the fact there the synthesis itself does not require special conditions and is relatively easy to conduct, the described method also allows for the upscale production of bioresorbable polyacids.

The obtained copolymers are a very interesting biomaterial. Unfortunately, the field of potential biomedical applications is narrowed by relatively low molecular weights, probably due to the great difficulties in the synthesis of the MTC-COOH monomer with the very high required chemical purity, as well as the occurrence of secondary reactions, such as chain transfer and intramolecular transesterification. This is also evidenced by the fact that similar difficulties were also encountered during the preparation of MTC-COOH/lactide copolymers with the use of lipase [11], as well as others containing side carboxyl groups [12,13].

It seems that the obtained copolymers can be an interesting carrier of many biologically active substances and drugs (especially for use in dermatology and cosmetology). In our next planned publication, we will present a detailed description of the physicochemical properties of the copolymers, with particular emphasis on the possibility of controlling their hydrophilic–hydrophobic balance, the course of their hydrolytic degradation, their antibacterial activity, and their cytotoxicity, which will allow for a reliable assessment of the biomedical suitability of these copolymers.

## Data Availability

Most of the data were included in the publication. The other source data presented in this study are available on request from the corresponding author. The data are not publicly available due to the present lack of access to a trusted public depository.

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
