# Peer review of "Synthesis of Polyacids by Copolymerization of l-Lactide with MTC-COOH Using Zn[(acac)(L)H2O] Complex as an Initiator"

_polymers, 2022, doi:10.3390/polym14030503_

Round 1

Reviewer 1 Report

In this work, authors have reported on..

Following corrections are needed:

-Chemical structures  may be presented in coloured format

-Whats are the advantages of the work over the existing ones

-How easy is it to scale up the synthesis process 

-Glass transition temperature may be explained in more detail

-Introduction may be strengthened by citing relevant articles such as: 

Chemical Reviews 120 (17), 9304–9362(2020); Polymers 2022, 14(2), 232;         Vacuum 146, 655-663(2017)                           

Author Response

Reviewer 1.

Thank you for your in-depth review. In accordance with the suggestions, we have made appropriate corrections to the text.

Reviewer 1

-Chemical structures may be presented in coloured format

Answer;

As suggested, to the text we introduced revised reaction of all schemes as well as chemical formulas in colour format.

Reviewer 1

 - What’s are the advantages of the work over the existing ones

Answer;

The advantages of obtaining copolymers containing pendant carboxyl groups in a one-stage polymerization reaction compared to the two-stage method described previously are indisputable. We described it in the manuscript text, comparing both the course of the synthesis process and the properties of the copolymers obtained by both methods. So far, in a one-step process, in addition to the method described by us, polyestercarbonates containing side carboxyl groups were obtained only when using enzymes (lipase) as catalysts. As the reviewer rightly pointed out, the presented work does not compare the properties of the copolymers obtained during the described copolymerization of L-lactide and MTC-COOH with the participation of a zinc complex initiator and the method using lipase. We have added this brief comparison in the Conclusion section. By using coordination initiators of the type described in our work Zn [(acac) L (H2O)], a higher degree of conversion of both monomers (95-97% in the case of the complex) (72% - 92% in the case of catalysing the reaction with an enzyme) is obtained, as well as higher average molecular weight Mw, from 32,000 - 49,000 g / mol and 7,800 - 9,000 g / mol respectively. On the other hand, the copolymers obtained by the method described in our manuscript are characterized by the larger distribution of molecular weights - Đ (2.1-3.8 and 1.6-2.3 respectively).

Reviewer 1.

-How easy is it to scale up the synthesis process

Answer;

The discussed method using a zinc chelate complex as a copolymerization initiator, unlike the two-stage method, is relatively easy to implement with the upscale synthesis. The used initiator is stable under normal storage conditions, does not decompose under the reaction conditions, and is insensitive to traces of water and contact with air. This initiator is soluble in the monomers melt, which guarantees the homogeneity of the reaction mixture also in the case of carrying out the reaction in a large volume. The synthesis itself does not require special conditions and is relatively easy to conduct. A certain technological disadvantage may be only the relatively long polymerization time needed to obtain high monomer conversion (more than 72h). However, a major difficulty in the production of the described copolymer on an upscale is the complicated and not very economical method of synthesizing a cyclic carbonate with a carboxyl group. A brief note on this topic was introduced in the manuscript in the Conclusion section.

Reviewer 1.

-Glass transition temperature may be explained in more detail

Answer;

As you suggested, we extended the discussion on the presumed causes of observed changes in the glass transition temperature synthesized copolymers.

Reviewer1

-Introduction may be strengthened by citing relevant articles such as Chemical Reviews 120 (17), 9304–9362(2020); Polymers 2022, 14(2), 232; Vacuum 146, 655-663(2017)                          

Answer;

We have added the literature references and a short commentary.

Reviewer 2 Report

The work presents the results of research on the preparation of bioresorbable functional copolyestercarbonates containing side carboxyl groups. The basic final properties of the obtained copolymers and the microstructure of their chains were determined.  The investigation will be interesting in polymer field and the paper could be published after revision.

-Zn[(acac)(L)(H2O)] was selected as the initiator of the polymerization reaction. How the authors obtained that this initiator is the most suitable. Have they tried also other initiators for the polymerization ?

-The authors can prepare polymer with very different Tg and TM temperatures. Which materials would have the best practical applications in terms of the temperatures ?

-Could the authors demonstrate some practical applications of the developed polymers ?

-The obtained copolymers are a very interesting biomaterial. Unfortunately, the field of potential biomedical application is narrowed by relatively low molecular weights. How the authors consider a possible ways to obtain high molecular weight polymers from the studied monomers ?

Author Response

 Reviewer 2.

The work presents the results of research on the preparation of bioresorbable functional copolyestercarbonates containing side carboxyl groups. The basic final properties of the obtained copolymers and the microstructure of their chains were determined.  The investigation will be interesting in polymer field and the paper could be published after revision.

Thank you for your in-depth review of our work. In line with the sent suggestions, we have introduced appropriate corrections and additions, which we hope will make the manuscript more readable.

Reviewer 2

-Zn[(acac)(L)(H2O)] was selected as the initiator of the polymerization reaction. How the authors obtained that this initiator is the most suitable. Have they tried also other initiators for the polymerization?

During previous preliminary research on the usefulness of zinc, zirconium and other metals acetylacetonates as initiators of polymerization and copolymerization of lactides or cyclic aliphatic carbonates, we observed for many of them relatively high resistance to deactivation with organic acids. This suggested that these compounds could be an initiator of the polymerization of monomers containing active carboxyl groups. This thesis was confirmed during research on the mechanism of MTC-COOH polymerization and copolymerization initiated with zirconium (IV) acetylacetonate. The obtained results suggested that the complexes obtained as a result of the exchange of some acetylacetonate ligands by reaction with proton donor compounds (in the case of initiation ROP of lactides and lactones with use of acetylacetonates such reactions occur at an early stage of polymerization initiation) should be probably an even better initiator of such reactions. We initially tested zirconium, molybdenum and zinc compounds (the selection was related to the relatively low toxicity of these metal coordination complexes) obtained in the ligand exchange reaction of appropriate acetylacetonates by the Schiff bases - synthesised during the condensation of amino acids with the aldehyde. The initiator used in the copolymerizations described in the presented manuscript was selected as the optimal due to the presented relatively high activity in ROP of lactide, good antibacterial properties and very low cytotoxicity. Among the investigated initiators, in the copolymerization of L-lactide with MTC-COOH, similar results were obtained in initiating reactions with zircon (IV) acetylacetonate and its derivatives obtained in the ligand exchange reaction of one acetylacetonate ligand in reaction with Schiff’s bases. This brief explanation has been included in the manuscript (part Introduction).

Reviewer 2

-The authors can prepare polymer with very different Tg and Tm temperatures. Which materials would have the best practical applications in terms of the temperatures?

Answer;

It seems that due to the planned application, copolymers with a glass transition temperature slightly higher than the temperature of the human body (containing 30-50% MTC-COOH units) seem to be optimal as a carrier of biologically active substances. These copolymers should be capable of easily forming micro-  and nanoparticles while still exhibiting some plasticity at body temperature.

Reviewer 2

-Could the authors demonstrate some practical applications of the developed polymers?

Answer;

As we wrote in the manuscript, the obtained copolymers can be used mainly as carriers of biologically active substances. These substances can be dispersed (or dissolved) in the polymeric material or linked by a covalent bond to a carboxyl group. The results of started research on the physicochemical properties of these copolymers will decide on the choice of the optimal method of the selected bioagent delivery with the help of the carrier formed with this material. Currently, using the good hydrophilicity of the obtained copolymers and the known antibacterial properties of polyacids, we test the obtained copolymers as a material for the formation of bioresorbable matrices (microspheres, nanospheres) releasing selected bioactive oils or septic substances useful for the formulation of creams and emulsions for use in cosmetology and dermatology.

The described copolymers can also find application as plasticizing, hydrophilic modifiers of other biodegradable polymers, for example by producing blends with modified polysaccharides.

Reviewer 2

-The obtained copolymers are a very interesting biomaterial. Unfortunately, the field of potential biomedical application is narrowed by relatively low molecular weights. How the authors consider a possible way to obtain high molecular weight polymers from the studied monomers?

Answer;

This is a question that is difficult to answer unequivocally. At present, one can only speculate as to the causes of this phenomenon.

It seems that mainly due to the great difficulties with the synthesis of monomer with the required very high chemical purity, as well as the occurrence of secondary reactions, chain transfer and intramolecular transesterification, obtaining a copolymer with high molecular weight is rather impossible, regardless of the chosen route (one or two-stage synthesis). This is also evidenced by the fact that similar difficulties were also encountered during the preparation of MTC-COOH/lactide copolymers with the use of lipase (where average molecular weights are even lower than those obtained in the method described in the work), or other copolymers containing side carboxyl groups with using the ROP mechanism (Journal of Polymer Science: Part A: Polymer Chemistry, Vol. 42, 2303–2312 (2004)). A solution that should increase the value of average molecular weights seems to be the development of a modified method of synthesis of MTC-COOH monomer used so that the product does not contain traces of carboxylic acid derivatives, as well as the search for a different type of initiator. Another solution is the possibility of coupling the chains of the synthesized copolymers by reaction of the hydroxyl end groups of the copolymer chains with diisocyanates.

Round 2

Reviewer 2 Report

If editor and other reviewers agree I recommend the paper for publication after the revision.